# Classification of EEG Signals Based on Sparrow Search Algorithm-Deep Belief Network for Brain-Computer Interface

**DOI:** 10.3390/bioengineering11010030

**Published:** 2023-12-27

**Authors:** Shuai Wang, Zhiguo Luo, Shaokai Zhao, Qilong Zhang, Guangrong Liu, Dongyue Wu, Erwei Yin, Chao Chen

**Affiliations:** 1School of Electrical Engineering and Automation, Tianjin University of Technology, Tianjin 300380, China; tjwangshuai1234@163.com (S.W.); 18856485920@163.com (Q.Z.); 984187431@163.com (G.L.); tsam0925@163.com (D.W.); 2Defense Innovation Institute, Academy of Military Sciences (AMS), Beijing 100071, China; zhiguo_luo_nudt@foxmail.com (Z.L.); lnkzsk@126.com (S.Z.)

**Keywords:** brain-computer interface, motor imagery, empirical mode decomposition, sparrow search algorithm, deep belief network

## Abstract

In brain-computer interface (BCI) systems, challenges are presented by the recognition of motor imagery (MI) brain signals. Established recognition approaches have achieved favorable performance from patterns like SSVEP, AEP, and P300, whereas the classification methods for MI need to be improved. Hence, seeking a classification method that exhibits high accuracy and robustness for application in MI-BCI systems is essential. In this study, the Sparrow search algorithm (SSA)-optimized Deep Belief Network (DBN), called SSA-DBN, is designed to recognize the EEG features extracted by the Empirical Mode Decomposition (EMD). The performance of the DBN is enhanced by the optimized hyper-parameters obtained through the SSA. Our method’s efficacy was tested on three datasets: two public and one private. Results indicate a relatively high accuracy rate, outperforming three baseline methods. Specifically, on the private dataset, our approach achieved an accuracy of 87.83%, marking a significant 10.38% improvement over the standard DBN algorithm. For the BCI IV 2a dataset, we recorded an accuracy of 86.14%, surpassing the DBN algorithm by 9.33%. In the SMR-BCI dataset, our method attained a classification accuracy of 87.21%, which is 5.57% higher than that of the conventional DBN algorithm. This study demonstrates enhanced classification capabilities in MI-BCI, potentially contributing to advancements in the field of BCI.

## 1. Introduction

Recent advancements in brain-computer interface (BCI) research have significantly impacted the field of artificial intelligence, introducing effective new research methodologies. BCI systems utilize brain signals to control external devices, facilitating direct human-computer communication and interaction [1]. These systems have been successfully applied in various domains. By decoding brain signals from the cerebral cortex, BCIs offer insights into an individual’s intentions and enable a variety of operations on external devices. In medical settings, BCIs are increasingly recognized for their potential to assist patients with neural impairments to regain autonomous movement. Specifically, Motor Imagery (MI)-based BCIs have demonstrated promising results in the rehabilitation of stroke patients [2]. This breakthrough has the potential to significantly improve the quality of life for individuals with disabilities. In addition, brain-machine interfaces have contributed significantly to sleep research [3]. The progression in BCI technology not only furthers artificial intelligence but also shows immense potential for improving healthcare and fostering human-machine integration [4]. MI facilitates structural and functional reorganization in stroke rehabilitation [5], achieved through the repeated activation of motor-neuron circuits. This process repairs connections between damaged neurons via neural plasticity, leading to improvements in motor dysfunction. Research into the recognition capabilities of brainwave signals is not only crucial for understanding brain behavior but also instrumental in advancing the medical and health industries [6], underscoring its importance for future technological developments.

The electroencephalogram (EEG), a conventional technique for capturing temporal fluctuations in brain activity, faces significant challenges regarding the quality of its signals. These EEG signals are often marred by various artifacts, both physiological and non-physiological. A critical step in EEG analysis is the removal of baseline power [7]. This process is not only fundamental but also essential in the feature processing of EEG signals. Researchers have developed a method known as InvBase for baseline correction. Experimental evidence has shown that this method effectively improves classification accuracy [8].

In the array of BCI paradigms [9], MI-based BCI is distinguished as a crucial subset within this field. MI-BCI primarily focuses on the identification and analysis of sensorimotor rhythm (SMR) signals [10] to decode cerebral signals into actionable commands for machines [11]. In the foundational studies of MI, the concept of Event-Related Desynchronization/Synchronization (ERD/ERS) was introduced by the BCI Laboratory at the Graz University of Technology in Austria. This concept serves as a key framework for differentiating MI tasks [12]. MI is conceptualized as a cognitive process in the human brain, not culminating in physical actions [13]. MI-BCI represents a leap beyond conventional paradigms, establishing a communication link between humans and machines. This connection enables cerebral control over devices such as wheelchairs and exoskeletons. It also advances neural rehabilitation based on neuroplasticity theories [14]. These developments position MI-BCI as a powerful tool in neural rehabilitation [15]. Early results have shown promising potential for the application of MI-BCI in stroke rehabilitation therapies [16]. A significant challenge in the BCI field lies in the enhancement of motor imagery classification and recognition. Addressing the need for more effective classification algorithms is crucial. Improving recognition accuracy in MI-based BCIs is especially vital in the context of medical research [17].

Numerous scholars have delved into research to enhance accuracy in this domain. Tong Guofeng and colleagues utilized a Deep Belief Network (DBN) model for classifying hyperspectral remote sensing images. They refined the training process of the DBN and validated their approach with the Salinas hyperspectral remote sensing image dataset [18]. This method demonstrated a notable improvement in classification accuracy compared to traditional approaches. Kamada S. and Ichimura T. also applied the DBN model, this time for video information classification. They adopted an adaptive learning method to determine the optimal network structure during training [19], achieving satisfactory levels of accuracy. Various EEG processing methods based on ERD and ERS have been proposed, including Common Spatial Patterns (CSP) [20], autoregressive models, wavelet transforms, and DBN. With the advancement in deep learning research, DBN has emerged as an effective technique for EEG signal processing. Numerous derivative methods based on DBN have been developed, such as sparse-DBN [21] and the Frequency-Domain Deep Belief Network (FDBN). The multiband FDBN, in particular, has shown an average accuracy improvement compared to the FDBN algorithm [22]. This advancement suggests the potential for optimizing DBN using the sparrow search algorithm (SSA). Furthermore, DBNs have been extensively applied across various fields. While they exhibit strong classification capabilities in these applications, DBNs often lack effective optimization strategies and are susceptible to becoming trapped in local optima, which can indirectly impact classifier performance [23]. There is a pressing need for an enhanced search algorithm to fine-tune the hyperparameters of DBN for superior classification performance.

Recent research developments, notably in 2020, saw Xue et al. introduce a new optimization algorithm known as the Sparrow Search Algorithm (SSA) [24]. This algorithm is distinguished by its robust optimization capability and rapid convergence speed [25,26]. Yin implemented SSA to optimize kernel coefficients and penalty coefficients for SVM classifiers, comparing the results with those from three traditional classifiers, namely CNN and SVM. The SSA-SVM classifier exhibited significant improvements overall [27]. The SSA algorithm, with its dual strengths in global and local search capabilities, effectively determines the hyperparameters of DBN, thus addressing DBN networks’ limitations. By optimizing DBN networks with SSA, we can enhance the classifier’s recognition ability. This approach has opened new avenues in the field of MI-BCI.

The primary contributions to this paper are as follows:We introduce a Sparrow Search Algorithm-based optimization for a Deep Belief Network model, specifically for classifying EEG data, providing a novel approach to EEG signal classification.To tackle the challenge of poor stability in motor imagery EEG data quality, our study integrates motor observation with motor imagery. This novel experimental paradigm enhances the quality of the EEG data collected for the experiment.The article employed various classification methods besides the SSA-DBN algorithm, namely DBN, GA-DBN, PSO-CNN, and PSO-DBN. Upon comparing the classification results of these algorithms, it was observed that the SSA-DBN algorithm exhibits superior classification capabilities.

## 2. Related Work

In this study, we employ the SSA-DBN approach to develop a framework for recognizing MI in EEG signals. Initially, we selected one self-collected dataset and two publicly available datasets, namely BCI IV 2a and SMR-BCI, to assess the classifier’s performance. During the data processing phase, we utilized EMD in tandem with the Hilbert-Huang Transform (HHT) for feature extraction from EEG signals. This method effectively captures the ERD and ERS phenomena, crucial markers of MI tasks. The application of SSA enabled us to fine-tune the DBNs hyperparameters, leading to enhanced recognition capabilities. Our approach demonstrated impressive performance not only on our private dataset but also on the BCI IV 2a and SMR-BCI public datasets. Relative to the unoptimized DBN network, we observed a 10.38% improvement in classification accuracy on our self-collected dataset. For the BCI IV 2a and SMR-BCI datasets, the accuracy enhancements were 9.33% and 5.57%, respectively. Overall, our proposed algorithm offers a new perspective on MI study, with the potential to contribute significantly to advancements in BCI technology.

The remainder of this paper is organized as follows: Section 3 details the materials and methods. Section 4 details the results. Section 5 and Section 6 present the discussion and conclude, respectively. This paper Limitations and Futures, with Section 7.

## 3. Materials and Methods

### 3.1. Dataset

#### 3.1.1. Experimental Design and Experimental Paradigm

Two computers are involved in the experimental setup: one for the participant (PC1) and one for the experimenter (PC2). Unity 3D software is equipped on PC1, through which the experimental paradigm is designed and developed. Instructions from PC2 are received by PC1 via the TCP communication protocol, and the starting time point are recorded for subsequent data alignment. MATLAB is installed on PC2, through which time alignment is conducted and the start signal is sent Real-time EEG data from the participant are acquired and stored utilizing the Curry8(8.0.3) software [28].

For the acquisition of EEG data, the international 10–20 system standard is adhered to with the employment of a 32-channel EEG cap. The EEG signals are relayed using a portable EEG signal amplifier. Through hardware connections, the EEG cap is linked to the amplifier. The framework of the experimental design is graphically represented in Figure 1, outlining the EEG signal acquisition system.

#### 3.1.2. Experimental Paradigm

Based on the MI-BCI system, the experimental paradigm for this study is crafted, and the action observation (AO) state is integrated. Each procedure is segmented into four distinct states: 3 s dedicated to AO, 3 s for MI, a 2-s intermission for rest, and a cue phase lasting 2 s. Comprising these four states is a single cycle, and eight such cycles make up each task. A visual representation of this paradigm is provided in Figure 2.

#### 3.1.3. Data Preprocessing

The EEG data of eight subjects were collected in this experiment, and all experiments were completed in the laboratory of Tianjin University. The experiment was approved by the Ethics Committee of Tianjin University (TJUE-2021-138). Original EEG signals are intrinsically faint, making the experimental data collection prone to artifacts. To counteract this vulnerability, a series of preprocessing measures are adopted. Initially, the gathered data are subjected to independent component analysis (ICA) to eliminate artifacts [29]. In our study, the EEG data undergoes a downsampling process to achieve a resolution of 512 Hz. Subsequently, it is filtered using a Butterworth filter within the 0.5–40 Hz frequency range. This filtering technique is crucial for effectively removing noise from brainwave signals [30]. The EEG guides this selection of attributes pertinent to MI-AO. Post-preprocessing, the EEG datasets are segmented. The central focus of this research revolves around the analysis of EEG data under two conditions: observation and imagination. Furthermore, temporal markers for each experimental condition are precisely set by the experimental paradigm. Valid data segments are identified as spanning 1 s before and 3 s after each condition. The data collected in the first second before the commencement of the experiment are utilized for baseline correction. This step is essential to reduce or eliminate any drift in the EEG signals. The initial three seconds of the experiment capture the pertinent task segment, as defined by our experimental paradigm. Each data acquisition encompasses 35 channels, a procedure reiterated eight times, culminating in a comprehensive total of 35 × 512 × 4 × 8 sampling points per experiment.

### 3.2. Public Datasets

#### 3.2.1. SMR-BCI Dataset

The Korea University has furnished a motion imagery SMR-BCI dataset [31], encompassing EEG data from a cohort of 14 subjects, delineated as S01 through S14. This dataset is bifurcated into two distinct MI task categories: the imagined movement of the right hand and the conjoint imagined movement of both feet. The EEG data for each participant are systematically divided into a training set and a testing set. The training compilation is composed of 100 MI trials, averaging 50 trials for each MI task delineation. Concurrently, the testing ensemble comprises 60 MI trials, averaging 30 trials per MI task category.

#### 3.2.2. BCI Competition IV Dataset 2a

In this investigation, the BCI Competition IV Dataset 2a (BCI IV 2a) [32] was employed for validation. The BCI IV 2a dataset, adhering to the international standard 10–20 system, encompasses EEG signals associated with MI and is derived from 9 subjects. Four distinct motor states, namely left hand, right hand, foot, and tongue, are represented, and these are aligned with the categories 0, 1, 2, and 3, respectively. Within the BCI IV 2a dataset, data from each subject are composed of signals emanating from two experimental rounds. Within each round, six experimental sets are included, with each set encompassing 48 MI state trials. As a result, comprehensive 288 MI trials are presented in the BCI IV 2a dataset for analysis and performance classification evaluation.

### 3.3. Method

#### 3.3.1. Sparrow Search Algorithm

Xue and Shen introduced a SSA that mimics the foraging behavior of sparrows [24]. This algorithm is characterized by its simplicity in principle, minimal parameter adjustments, and ease of programming. In comparison to Particle Swarm Optimization (PSO) [33] and Genetic Algorithm (GA) [34] in terms of function optimization, SSA demonstrates superior search capabilities. One notable advantage of the Sparrow Search Algorithm is its relatively low temporal and spatial complexity [35], making it well-suited for addressing large-scale problems. The spatial complexity of SSA is primarily influenced by the search space size, usually requiring modest memory resources. However, the temporal complexity of the algorithm may escalate when dealing with extensive problems, depending on the breadth and depth of the search space. The procedural steps of the algorithm are illustrated in Figure 3.

The computational trajectory of the algorithm unfolds as follows [27]:

First, represent a population of n Sparrows as n×d -d-dimensional vectors using matrices,
(1)$$X=\left[\begin{array}{cccc} X_{1,1} & X_{1,2} & \cdots & X_{1,d}\\ X_{2,1} & X_{2,2} & \cdots & X_{2,d}\\ \vdots & \vdots & \ddots & \vdots \\ X_{n,1} & X_{n,2} & \cdots & X_{n,d} \end{array}\right]$$

In the above equation, $X_{ij}$ represents the position of the $j_{th}$ dimension of the $i_{th}$ sparrow; The fitness of the sparrow population can be represented as:(2)$$F_{X}=\left[\begin{array}{cccc} f((X_{1,1} & X_{1,2} & \cdots & X_{1,d}))\\ f((X_{2,1} & X_{2,2} & \cdots & X_{2,d}))\\ \vdots & \vdots & \vdots & \vdots \\ f((X_{n,1} & X_{n,2} & \cdots & X_{n,d})) \end{array}\right]$$

In the above equation, f is the fitness of an individual sparrow.
(3)$$X_{i,j}^{t+1}=\left\{\begin{array}{c} X_{i,j}.\exp\left(-\frac{i}{\alpha .{iter}_{max}}\right),R_{2}<ST\\ X_{i,j}+Q.L,R_{2}\geq ST \end{array}\right.$$

$X_{ij}$ represents the regional data of the $i_{th}$ Sparrow in the $j_{th}$ dimension. $\alpha \left(\alpha \in \left(0,1\right]\right)\;$represents a randomly distributed number. $R_{2}(R_{2}\in [0,1])$ and ST(ST∈[0.5,1]) represent the indicative number and the security number, respectively. Q represents a symbol conforming to a normal distribution. When $X_{ij}$ is arriving at the position of the joiner. The updated formula is as follows:(4)$$X_{i,j}^{t+1}=\left\{\begin{array}{c} Q.\exp\left(\frac{X_{worst}-X_{i,j}^{t}}{i^{2}}\right),i>n/2\\ X_{P}^{t+1}+\left|X_{i,j}-X_{P}^{t+1}\right|.A^{+}.L,otherwise \end{array}\right.$$

Follows A is a 1×d matrix, and $A^{+}=A^{T}(AA^{T}{)}^{-1}.$ When $X_{ij}$ is in a state of surveillance and early warning behavior. The updated formula is as follows:(5)$$X_{i,j}^{t+1}=\left\{\begin{array}{c} X_{best}^{t}+\beta .\left|X_{i,j}^{t}-X_{best}^{t}\right|,f_{i}>f_{g}\\ X_{i,j}^{t}+K.\left(\frac{\left|X_{i,j}^{t}-X_{worst}^{t}\right|}{\left(f_{i}-f_{w}\right)+\varepsilon }\right),f_{i}=f_{g} \end{array}\right.$$
where $f_{g}$ and $f_{w}$ represent the global best fitness value and the global worst fitness value, respectively.

#### 3.3.2. Deep Belief Networks

DBN epitomizes a class of neural networks adept at executing both supervised and unsupervised learning paradigms. The distinctive architecture of the DBN is delineated in the subsequent diagram. This network is an amalgam of multiple strata of Restricted Boltzmann Machines (RBM), crowned by a solitary layer of backpropagation (BP) networks. The RBM, a bidirectional, fully connected neural network structure, is composed of visible and hidden layers, serving as a linchpin in optimizing the overarching DBN network, the DBN network structure is shown in Figure 4. Upon fine-tuning the RBMs, the DBN is poised to distill high-level features [36]. 

In the unsupervised training scenario, it is crucial to ensure that the feature vectors of the trained RBM network can be mapped to different feature spaces while retaining their informative characteristics. The DBN network proceeds to extract the feature information layer by layer, with the BP network designated in the last layer to retrieve the output feature vectors from RBM for supervised classification training. As illustrated in the diagram, it is apparent that although each RBM network operates independently and does not engage in the mapping of feature vectors from other RBM network layers, the training regimen of RBM facilitates the initialization of weight parameters for each deep layer of the BP network. This assists the network in averting the challenges of local optima and extended training durations that may arise due to the random initialization of weight parameters in the BP network.

The principle of the DBN network is delineated as follows: During supervised fine-tuning training, forward propagation is carried out to obtain specific output values from the input. Thereafter, backward adjustment is applied to propagate and update the weight values W and bias terms b of the network. During forward propagation, pre-trained weights W and biases b are utilized to compute the activation values of each hidden neuron.
(6)$$h^{\left(l\right)}=W^{\left(l\right)}\cdot v+b^{\left(l\right)}$$
where l is the index value of the layer in the neural network, and the values of W and b are as follows:(7)$$W=\left[\begin{array}{cccc} W_{1,1} & W_{2,1} & \cdots & W_{m,1}\\ W_{1,2} & W_{2,2} & \cdots & W_{m,2}\\ \vdots & \vdots & \ddots & \vdots \\ W_{1,n} & W_{2,n} & \cdots & W_{m,n} \end{array}\right],b=\left[\begin{array}{c} b_{1}\\ b_{2}\\ \vdots \\ b_{n} \end{array}\right]$$
where $W_{i,j}$ represents the weights, and then the activation values of each hidden neuron in each layer are calculated. Afterwards, they are normalized.
(8)$$\sigma (h_{j}{)}^{\left(l\right)}=\frac{1}{1+e^{-h_{j}}}$$

Finally, we obtain the activation function $f\left(\cdot \right)\;$ and output values X of the output layer.
(9)$$h^{\left(l\right)}=W^{\left(l\right)}\cdot h^{\left(l-1\right)}+b^{\left(l\right)},X=f\left(h^{\left(l\right)}\right)$$

In backward propagation, the backpropagation algorithm is used with the mean squared error criterion to update the parameters of the entire network. The cost function is as follows:(10)$$E=\frac{1}{N}\sum_{i=1}^{N}[{\hat{X}}_{i}(W^{l},b^{l})-X_{i}{]}^{2}$$
where, E is the squared error of DBN learning, where ${\hat{X}}_{i}$ and $X_{i}\;$ represent the output and ideal output of the output layer, respectively. i is the sample index, and $\left(W^{l},b^{l}\right)$ represents the parameters of the weights and biases to be learned in layer l.

The gradient descent method is used to update the weights and biases of the network, where λ represents the learning rate.
(11)$$\left(W^{l},b^{l}\right)\leftarrow \left(W^{l},b^{l}\right)-\lambda \cdot \frac{\partial E}{\partial \left(W^{l},b^{l}\right)}$$

Based on the above discussion, by selecting appropriate numbers of hidden layers, determining the number of neurons in each layer, and setting the learning rate, followed by iterating through a specified number of times during training, we can achieve the desired network mapping model.

#### 3.3.3. The SSA-DBN Method

DBNs have manifested encouraging performance in the realms of feature extraction and classification. Nevertheless, the optimal network structure of DBNs is pivotal for their performance. To tackle this, the SSA can be harnessed to optimize the DBNs. SSA identifies the Sparrow individual exhibiting the highest fitness, symbolizing the optimal position of Sparrow, to configure the optimal network structure of the DBN, culminating in the SSA-DBN classification recognition model.

The SSA-DBN model is bifurcated into two modules: the data preprocessing module and the SSA-DBN module. The data preprocessing module readies the raw data for input into the SSA-DBN module. The SSA-DBN module executes the classification and recognition of the input data. This entails training the module with the training dataset to optimize its parameters and achieve the optimal classification recognition model. Subsequently, the test dataset is channeled into the trained SSA-DBN module to obtain classification outcomes.

The algorithm unfolds as follows: Initially, a specific number of Sparrow populations are randomly initialized, and relevant parameters are established. The fitness function F(X) is evaluated to identify the optimal value. Subsequently, the positions of the Sparrows are updated, and the fitness function’s optimal value is recalculated. If the discerned optimal value aligns with the global optimum, the parameter X corresponds to the optimal network structure of DBN. Otherwise, the updating of the Sparrows’ positions continues until either the optimal value is attained or the maximum iteration limit is reached. Ultimately, based on the obtained parameters, the optimal classification model is constructed for recognition and classification tasks. The SSA algorithm boasts a higher convergence speed, augmented accuracy, and a lower propensity for entrapment in local optima, showcasing a global characteristic. Employing SSA to optimize DBN guides the Sparrow to the optimal position, as depicted in Figure 5 of the model.

### 3.4. Feature Extraction: Empirical Mode Decomposition and the Hilbert–Huang Transform

The technique of EMD is extensively utilized for the analysis of nonlinear and non-stationary signals, exhibiting utility in the extraction of features from such signals. EMD operates by decomposing a signal into Intrinsic Mode Functions (IMFs), which epitomize unique patterns inherent in the data. IMFs render valuable insights into the frequency range characteristics of the original EEG signal, thereby enabling effective feature extraction [37].

Nevertheless, EMD may be plagued by issues such as mode mixing and the presence of residual white noise in the intrinsic mode components. To ameliorate these challenges, a fully adaptive noise Complete Ensemble Empirical Mode Decomposition (CEEMD) method is proposed. The decomposition procedure encompasses the following steps:(12)$$x_{i}\left(t\right)=x\left(t\right)+\varepsilon \delta_{i}\left(t\right)$$

In the above equation, $x\left(t\right)$ is the signal to be decomposed; ε is the Gaussian white noise weighting coefficients; i = 1, 2, 3…; $\delta_{i}\left(t\right)$ is the Gaussian white noise generated during the $i_{th}$ iteration. Perform EMD decomposition on the sequence $x_{i}(t)$.
(13)$$IMF_{1}(t)=\frac{1}{K}\sum_{i=1}^{K}IMF_{1}^{i}\left(t\right)\;\;$$
(14)$$r_{1}\left(t\right)=x\left(t\right)-IMF_{1}\left(t\right)$$

In the above equation, $IMF_{1}(t)$ represents the first mode component; $r_{1}(t)$ represents the Residual signal after the first decomposition.
(15)$$\;IMF_{j}(t)=\frac{1}{K}\sum_{i=1}^{K}E_{1}\left(r_{j-1}\left(t\right)+\varepsilon_{j-1}E_{j-1}\left(\delta_{i}\left(t\right)\right)\right)\;\;$$
(16)$${\;r}_{j}\left(t\right)=r_{j-1}\left(t\right)-IMF_{j}\left(t\right)$$

In the equation, $IMF_{j}(t)$ denotes the component of the $j_{th}$ order mode procured post-CEEMDAN decomposition. This procedure is reiterated until one of the following conditions emerges, thereby concluding the EMD decomposition process:

Achievement of the $j_{th}$ order IMF component-$IMF_{j}(t)$; The magnitude of the residual signal, termed $r_{j}(t)$, falls below a predefined threshold. The residual signal, denoted as $r_{j}(t)$, manifests as either a monotonic function or a constant.

The HHT serves to transform the foundational elements of a signal, typically sinusoidal signals, into Intrinsic Mode Functions (IMFs). From a theoretical perspective, HHT lays the groundwork for the analysis of oscillating frequencies inherent to non-stationary signals. Through the application of the Hilbert transform to every mode component that fulfills the IMF criteria, one can deduce the corresponding instantaneous frequency along with the instantaneous amplitude of the IMF.
(17)$$H\left[c_{j}\left(t\right)\right]=\frac{1}{\pi }\int_{-\infty }^{+\infty }\frac{c_{j}\left(t\right)}{t-\tau }d\tau$$
(18)$$A\left[c_{j}\left(t\right)\right]=c_{j}\left(t\right)+iH\left[c_{j}\left(t\right)\right]=a_{j}\left(t\right)e^{i\theta_{j}\left(t\right)}$$
where $c_{j}(t)$ is the mode component satisfying the IMF condition. The corresponding instantaneous frequency is represented as follows:(19)$$f_{j}\left(t\right)=\frac{1}{2\pi }\frac{d\theta_{j}\left(t\right)}{dt}\;$$

Therefore, we can obtain:(20)  cj(t)=Re⁡[aj(t)eiθjt]=Reaj(t)exp⁡[i2π∫fjtdt]

By the formula of EMD, we can derive:(21) xt≈Re⁡∑j=1Naj(t)exp⁡[i2π∫fjtdt]

By using the above equation, we can obtain the Hilbert HHT time-frequency spectrogram as follows:(22)$$H(\omega ,t)=\sum_{j=1}^{N}b_{j}a_{j}\left(t\right)\;$$

Further, we can obtain the Hilbert marginal spectrum:(23)$$h\left(\omega \right)=\int_{0}^{T}H\left(\omega ,t\right)dt$$

In this research, the marginal spectrum for each state is analyzed with a window length and shift of 1 Hz. This analysis yields 30 distinct feature vectors per state.

### 3.5. Parameter Optimization for Classification

To enhance the performance of the DBN network within the SSA-DBN model, the structural parameters of DBN are optimized using the SSA method, and specific parameters are established. The specific parameters of the SSA iteration process are shown in Table 1.

Drawing from the SSA optimization procedure, we identify the peak fitness function F(X), facilitating the determination of the DBNs structural parameters, denoted as X. Subsequently, the DBN is formulated, leveraging the optimization outcomes from the SSA to ascertain the optimal neuron count, termed Best_pos. Within this framework, the implications of diverse output parameters are delineated in Table 2.

In the present research, a three-layered RBM model of the DBN is employed for the extraction of features. Initially, the neuron count in each network layer is established as Best_pos (1,1), Best_pos (1,2), and Best_pos (1,3), in sequence. Following this, the BP classifier is utilized for the classification task. During the training phase of the RBMs, parameters are configured as follows: The number of iterations is 300, the batch of training samples is 30, and the learning rate factor is (1,4).

For the transition from DBN to a deep reverse fine-tuning network, the command “nn” = dbnunfoldtonn(DBN, i) is implemented, where i symbolizes the quartet of classification outcomes. These outcomes resonate with the four classification recognition outputs from the BCl IV 2a public dataset. During the reverse fine-tuning phase, the parameters are as follows: The number of reverse fine-tuning iterations is Best_pos (1,5), the learning rate factor is 0.001, the number of training samples per iteration is 30, and the activation function is SoftMax.

### 3.6. The Paired Samples t-Test

The paired sample t-test, a widely used statistical method [38], is employed in this study to conduct significance testing between the SSA-DBN classification algorithm and other competing classification algorithms. Specifically, we assess the performance of the SSA-DBN algorithm within the SMR-BCI dataset by comparing it against other algorithms. The fundamental steps for conducting a significance test between the SSA-DBN algorithm and the DBN algorithm include:

Proposing the Hypothesis H: Here, $H_{0}$ posits that no significant difference exists in the mean values of the two samples, while $H_{1}$ suggests a significant difference in these mean values.

Set the significance level p-value: In this verification, the significance level p-value is set at 0.05.

Calculate the statistical t-value:(24)$$t=\frac{{\bar{X}}_{1}-{\bar{X}}_{2}}{S_{w}\sqrt{\frac{1}{N_{1}}+\frac{1}{N_{2}}}}$$

In the above formula, $N_{1}$ and $N_{2}$ represent the sample size, $S_{w}$ represent the sample variance, and ${\bar{X}}_{1}$ and ${\bar{X}}_{2}$ represent the mean values of the samples.

Calculation of the t-test p-value: Based on the formula, the t-value is calculated. Assuming the null hypothesis H0 holds, the statistic conforms to a t-distribution with specific degrees of freedom. The p-value can be derived using a t-distribution table or corresponding functions. A p-value less than 0.05 leads to the rejection of H0, thereby supporting H1. This indicates a significant disparity in the mean values between the two methods.

## 4. Results

In this paper, data from eight subjects harvested from a self-collected dataset are meticulously processed through a series of preprocessing and feature extraction stages, employing the SSA-DBN methodology for classification.

For the scrutinization of an individual’s EEG signal, the signal undergoes decomposition facilitated by CEEMDAN (Complete Ensemble Empirical Mode Decomposition with Adaptive Noise) [39]. The resultant IMF components are illustrated in Figure 6, where each component correlates with specific frequency bands. Specifically, the IMF4-IMF5 components predominantly manifest frequency elements centered approximately at 0 Hz. Conversely, the frequency spectrum of MI-EEG signals conventionally aligns with the μ  and β  rhythms. Consequently, the terminal two components are disregarded.

Following the prior analysis, the first three components (IMF1–IMF3) were derived from the EMD decomposition and subjected to a Hilbert transformation to produce the frequency spectrum. The resulting three-dimensional frequency spectrum, however, was found unsuitable for feature extraction. Consequently, a marginal spectrum analysis was conducted, yielding a two-dimensional marginal spectrum better suited for feature extraction. With the implementation of this marginal spectrum analysis, the marginal spectrum for each time point was windowed, adopting a window length of 1 Hz and a shift of 1 Hz. A ceiling for the frequency range was established at 30 Hz. Through this methodology, 30 feature vectors were generated for each event, facilitating a reduction in data complexity and an enhancement in classification accuracy [40].

Following the description outlined, our feature analysis will concentrate on channels C3, C4, and CZ. These channels are located in the central region of the brain, an area recognized as crucial for motor imagery activities [41]. The subsequent figures show the analysis of the MI-AO state across channels C3, CZ, and C4 for letting go and handshakes:

An examination of the letting go state in Figure 7a reveals notable differences in the energy spectrum of MI and AO under the C3 channel, particularly around 10 Hz and 25 Hz. In contrast, under the CZ channel, AO and MI display minimal variance within the same frequency range. Meanwhile, the C4 channel shows a clear energy disparity near 10 Hz. This pattern is mirrored in the handshake state, as depicted in Figure 7b, where most energy differences are concentrated around 10 Hz and 25 Hz.

From the observations in Figure 7, it becomes apparent that distinct variations manifest in the alpha and beta frequency bands during motor imagery. The alpha band, typically ranging from 8 to 13 Hz, is predominant when a person is awake but relaxed. Conversely, the beta band, spanning 14 to 30 Hz, is commonly associated with normal cognitive activities. It is important to note that the Gamma frequency band, generally above 30 Hz, does not significantly relate to motor imagery activities [42]. In summary, a comprehensive analysis of Figure 8 demonstrates that the electroencephalographic characteristics of MI and AO exhibit substantial separability post-feature extraction via EMD + HHT. This finding provides robust data to support the subsequent application of the SSA-DBN classification algorithm.

Based on the above data, feature extraction is performed, and the extracted features are then classified and recognized using the method proposed in this paper and other methods. The three optimization methods are as follows: The Genetic Algorithm-based Deep Belief Network (GA-DBN) employs GA to optimize the structure and hyperparameters of the DBN [43]; the Particle Swarm Optimization-based Deep Belief Network (PSO-DBN) combines the PSO algorithm with the DBN to optimize the structure and parameters of the DBN [44]; and the Particle Swarm Optimization-based Convolutional Neural Network (PSO-CNN) is a CNN that utilizes PSO. This method employs the PSO algorithm to optimize the structure, weights, and hyperparameters of the CNN [45]. The ensuing classification outcomes undergo rigorous 5-fold cross-validation, with the computed mean value serving as the definitive measure of classification accuracy. Additionally, we have calculated the standard deviation and standard error of the data, which are supplemented in the table below. Additionally, we have computed the standard deviation and standard error for the data, which are detailed in the subsequent table. The inclusion of the standard deviation and standard error allows for a comprehensive comparison of the dispersion and variability among individual data samples relative to the overall dataset [46]. The comparative discernment of this methodology by others concerning classification recognition based on the self-collected data are elucidated in Table 3 herein.

After completing the analysis of the self-collected dataset, the same methodology was applied to two publicly available datasets for classification. The classification results are presented in Table 4 and Table 5.

The data presented in the three tables indicates that the SSA-DBN algorithm proposed in this study exhibits superior classification performance across all datasets. Figure 7 compares and analyzes the average classification accuracies of various algorithms across these datasets, with the paired t-test significance indicated within the graph.

Analysis of Figure 7 reveals that in binary datasets, specifically MI-AO and SMR-BCI, the SSA-DBN algorithm achieves a peak classification accuracy of 89.33%. Notably, within the BCI IV 2a dataset, which involves four categories, the highest average classification accuracy recorded is 86.14%. Across all three experimental datasets, SSA-DBN stands out as the superior classifier, exhibiting particularly remarkable performance in the self-collected dataset. This study also determined double-sided probability values (*p*-values) for the SSA-DBN classification method in comparison to other methods within the same dataset. The significance evaluation indicates substantial differences between various classification techniques and the SSA-DBN approach detailed in this paper.

## 5. Discussion

The primary objective of this research is to address the challenge of determining the hyperparameters for classifiers in EEG classification tasks. The SSA is adept at finding optimal local solutions. The DBN demonstrates exceptional efficacy in feature extraction, making it well-suited for feature classification [36]. Utilizing SSA to optimize the DBN network enables us to achieve optimal values, thus constructing an ideal classification model. The classification process yielded positive outcomes.

The burgeoning research in brain-machine interfaces offers new hope in rehabilitation therapies for individuals with disabilities. There is an increasing need for diverse, intelligent products in rehabilitation medicine for disabled people. The innovation and adoption of various smart products are progressively superseding traditional manual approaches [47], enhancing convenience for subsequent treatments. Elderly individuals with physical challenges stand to benefit significantly from smart technologies that facilitate the control of household appliances. This technological advancement represents a substantial benefit for people with disabilities [48]. It is crucial to analyze the brainwave data of stroke patients and correlate it with pertinent physiological responses, addressing a pressing contemporary need. This research extends hope to stroke patients [2].

Effective methods of signal analysis significantly aid in deciphering brain characteristics. In the initial phase of preprocessing EEG signals, artifacts crucially influence the imaging quality. The EEG signals are predominantly affected by physiological artifacts, including eye and muscle movements. Present-day algorithms for processing these artifacts encounter various challenges, with potential solutions discussed in reference [7]. Furthermore, the removal of baseline noise is a vital component of EEG signal preprocessing. The implementation of inverse filtering techniques for this purpose has yielded promising outcomes [8]. These scientific developments in both domains have laid the groundwork for future progress in pattern recognition.

In this study, we utilize EMD + HHT for feature extraction from EEG signals. As evidenced in Figure 7, it is prudent to retain IMF1~IMF3 for feature extraction. This approach not only reduces processing time, enhancing computational efficiency, but also lays the groundwork for subsequent classification tasks.

Upon applying this feature extraction method to three datasets, the classification results are depicted in Figure 8. Five classifiers were employed to assess the experimental performance. Among these, the SSA-DBN classifier consistently showed superior performance across all datasets, particularly excelling in the self-collected dataset with a higher accuracy rate of 0.62% compared to the public SMR-BCI dataset. Additionally, in the BCI IV 2a dataset, our method surpassed PSO-CNN by 5.69% [49]. Moreover, when compared to the DBN classifier without SSA optimization, the SSA-DBN network demonstrated enhanced classification performance in all three datasets.

While traditional DBN networks are known for their commendable classification capabilities, their limitations in hyperparameter determination restrict their overall efficacy [36]. Therefore, selecting an optimization method that boosts classification proficiency is crucial. This study undertakes comparative analyses with three alternative algorithms, ultimately identifying the SSA-DBN optimization algorithm to elevate classification performance [42,43,44]. Prior research predominantly focused on classification methods using various public datasets. This study expands upon existing literature by incorporating self-collected datasets. The contributions of this paper are twofold: enhancing classifier performance through SSA-DBN optimization and enriching existing research with additional datasets.

## 6. Limitations and Future

Limitations: While the feature extraction methodologies and classification recognition algorithms outlined in this study have yielded relatively positive outcomes, it is important to acknowledge certain limitations.

Firstly, the experimental data did not categorize subjects by gender, resulting in a skewed male-to-female ratio within the dataset. For future studies, ensuring gender balance among participants is essential to enhancing the broader applicability of the research findings.

Secondly, despite the proficiency of our feature extraction and classification methods in categorizing data, there is room for improvement in the data collection process. Efforts should be directed towards minimizing the impact of noise on experimental data. The challenge of eliminating wireless interference during experiments persists, which leads to the presence of artifacts in the EEG data.

Lastly, the time-intensive nature of EEG experiments can lead to decreased attention levels in participants during later stages. Additionally, a decline in the conductivity of electrode resistance over time can adversely affect the quality of research data.

Future Directions: The exploration of brain-computer interfaces holds considerable promise for clinical applications. Future research could involve collaborations with hospitals to collect and analyze EEG data from specific patient groups. Comparing this data with that of normal subjects will enable this study of differential patterns. Additionally, there is potential for integrating classification algorithms into online systems, which could significantly enhance the practical utility of brain-computer interfaces and foster the progression of clinical applications in this domain.

## 7. Conclusions

In this paper, we utilized the SSA-DBN classification model to conduct a comprehensive analysis of three datasets: the self-collected dataset, BCI IV 2a, and SMR-BCI. Initially, we optimized the paradigm of the self-collected data to alleviate the fatigue experienced by subjects during the observation of a single MI task. EEG signals were then collected from eight subjects and underwent preprocessing and EMD + HHT feature extraction. The extracted features were input into our SSA-DBN model, which had been optimized using the SSA. A feature extraction analysis was performed on the self-collected EEG signals, establishing a theoretical foundation for subsequent classification recognition. To evaluate the performance of our SSA-DBN algorithm, we conducted a comparative analysis with other classification algorithms, including DBN, PSO-CNN, PSO-DBN, and GA-DBN. Additionally, to ensure fairness, we validated our algorithms against other publicly available datasets, and various classification methodologies were applied to the self-collected dataset. The results obtained demonstrated that both the classification model and the self-collected dataset presented in this paper possess commendable generalization capabilities. These findings highlight the effectiveness and robustness of our proposed SSA-DBN algorithm for EEG classification tasks.

## Figures and Tables

**Figure 1 bioengineering-11-00030-f001:**
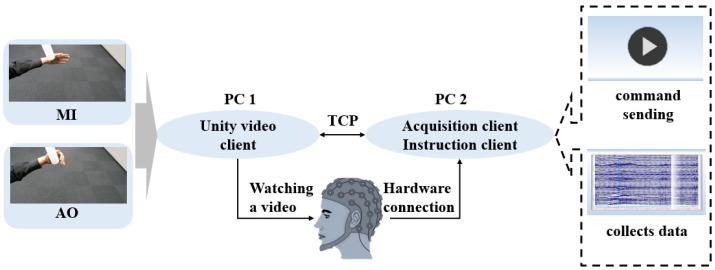
Brain Signal Acquisition System. The system consists of a main experimental terminal and a subject terminal, with the main terminal responsible for data collection and the subject terminal facilitating video induction. Subjects establish connections with both terminals through hardware.

**Figure 2 bioengineering-11-00030-f002:**
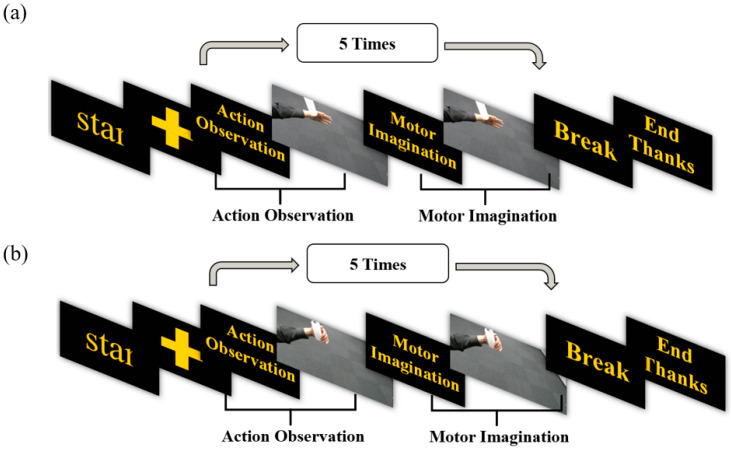
The experimental paradigm of the MI-AO dataset. (**a**) represents the experimental paradigm of action observation and motor imagination in the task of letting go of an object. (**b**) represents the experimental paradigm of action observation and motor imagination in the task of handshaking an object.

**Figure 3 bioengineering-11-00030-f003:**
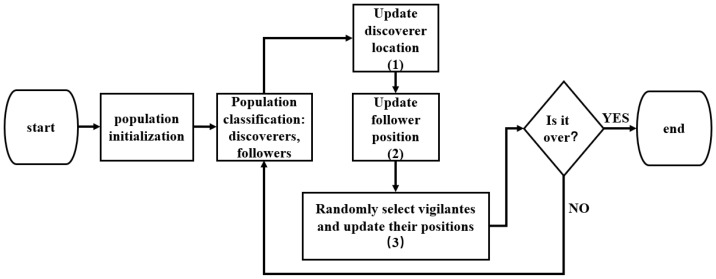
The flowchart of the sparrow search algorithm.

**Figure 4 bioengineering-11-00030-f004:**
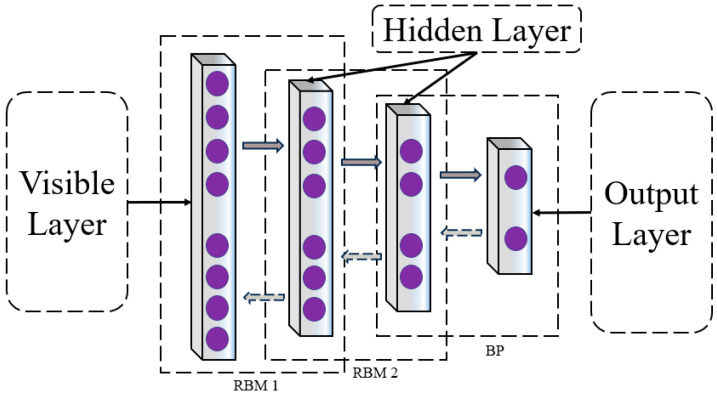
DBN network architecture diagram. The network consists of a visible layer, hidden layers, and an output layer, in which the hidden layers incorporate the RBM and BP networks for bidirectional connections.

**Figure 5 bioengineering-11-00030-f005:**
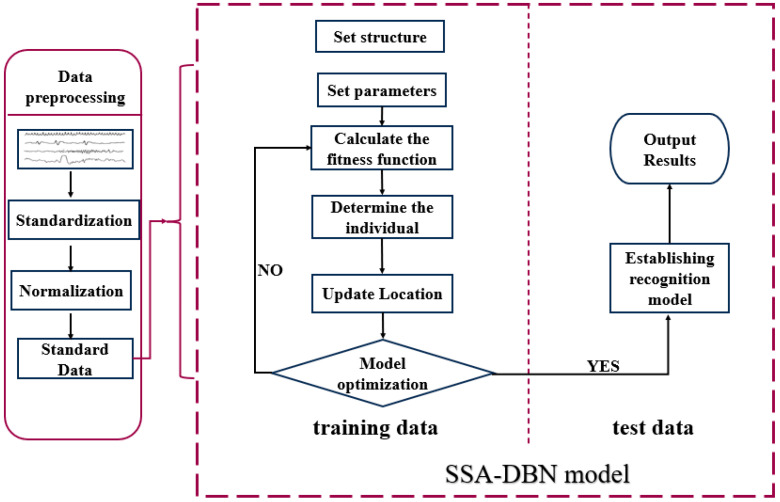
SSA-DBN classification and recognition model framework. The model undergoes data preprocessing before training and testing data within the SSA-DBN model.

**Figure 6 bioengineering-11-00030-f006:**
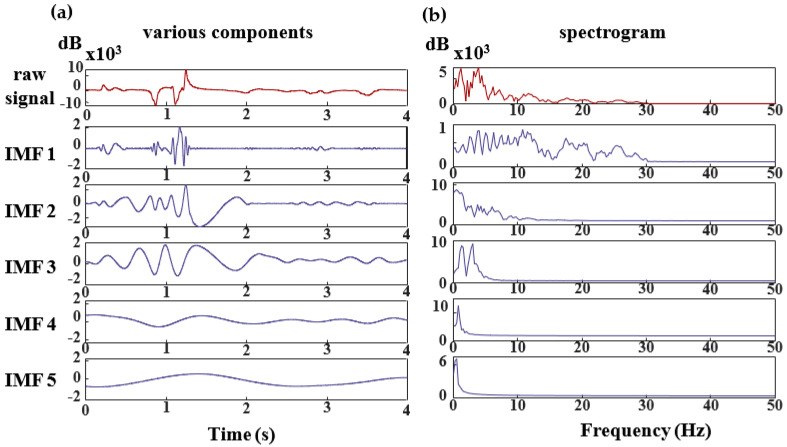
IMF component map of the EEG signal after Complete Ensemble Empirical Mode Decomposition with Adaptive Noise (CEEMDAN) decomposition. In Figure (**a**,**b**), the red color represents the original signal, while the purple color represents the various component signals. Figure (**c**) illustrates the instantaneous frequency and instantaneous amplitude of the three component signals.

**Figure 7 bioengineering-11-00030-f007:**
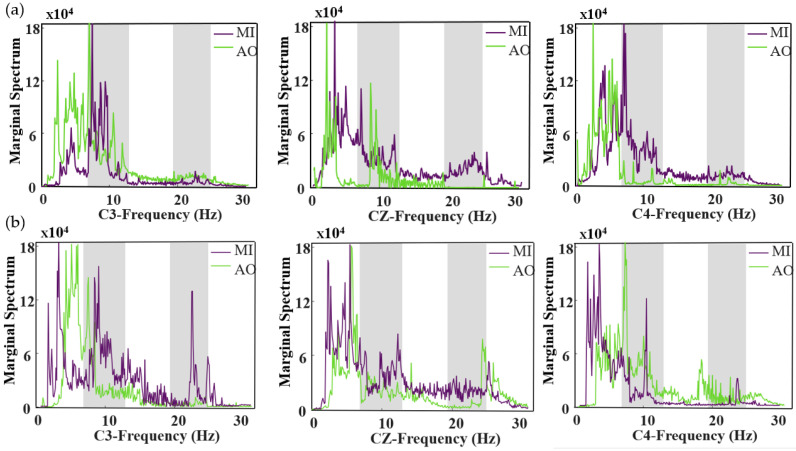
Marginal Spectrum Analysis of C3, CZ, and C4 across Different States in Letting Go and Handshake. (**a**) For the letting go state, the figure presents marginal spectral contrast diagrams showcasing action observation and motor imagery across the three channels. (**b**) For the handshake state, it illustrates marginal spectral contrast diagrams for action observation and motor imagery, again for the three channels.

**Figure 8 bioengineering-11-00030-f008:**
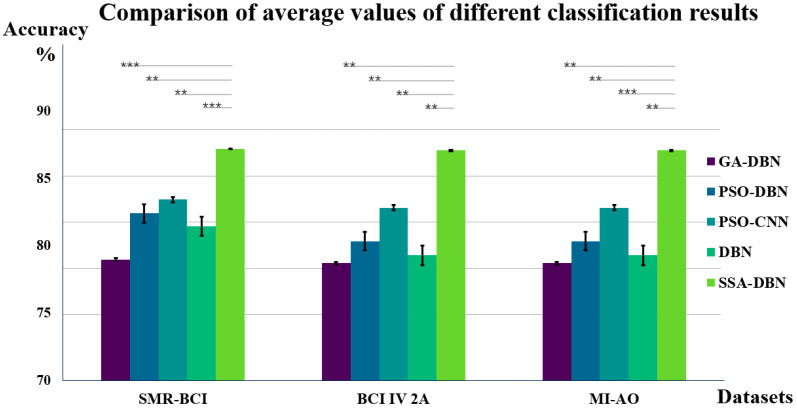
Classification Accuracy (%) of Different Algorithms. This figure encompasses three datasets and five classification methods. For each dataset, the accuracy of each classification algorithm is juxtaposed with the SSA-DBN algorithm’s accuracy via significance analysis. In the graph, the longest gray line signifies the algorithm most closely aligned with SSA-DBN, while the shortest line represents the classification method nearest to the DBN algorithm. *** denotes *p* < 0.001, ** denotes *p* < 0.05. All algorithms in the figure display significant differences compared to SSA-DBN. Among the five methods, SSA-DBN exhibits the smallest standard error.

**Table 1 bioengineering-11-00030-t001:** The specific parameters of the SSA iteration process.

Parameter Name	Value
Upper bounds	100.1, 100.2, 100, 0.1, 1500
Dimension of the independent variable	5
Lower bounds	5, 5, 5.2, 0.001, 300
Number of sparrows	5
Maximum number of iterations	20
Warning value and follower ratio	0.8, 0.2

**Table 2 bioengineering-11-00030-t002:** The implications of diverse output parameters.

Output Parameters	Related Definitions
Highest fitness function F(X)	Denotes the paramount solution procured via the SSA (Social Spider Algorithm) optimization
DBN structural parameters X	Alludes to the DBN network’s structural parameters ascertained from the peak fitness function
Optimal neurons count Best_pos	Signifies the optimal neuron count acquired through SSA optimization.

**Table 3 bioengineering-11-00030-t003:** Dataset classification accuracy (%) (AVG ± S.D).

	Proposed Method		Other Methods		
	SSA-DBN	GA-DBN	PSO-DBN	PSO-CNN	DBN
1	89.47	76.52	75.32	75.40	73.21
2	90.92	89.98	77.25	91.63	81.56
3	87.96	80.34	76.38	85.78	69.76
4	90.74	82.31	83.25	83.56	83.23
5	88.64	68.9	88.76	88.54	79.45
6	75.91	77.23	75.8	79.34	78.27
7	89.79	90.32	78.96	86.24	82.74
8	89.21	78.21	72.31	81.77	71.39
mean	**87.83**	**80.48**	**78.50**	**84.03**	**77.45**
std	**4.56**	**6.67**	**4.87**	**4.85**	**4.96**
SE	**1.53**	**2.22**	**1.62**	**1.61**	**1.65**

**Table 4 bioengineering-11-00030-t004:** BCI IV 2a dataset classification accuracy (%) (AVG ± S.D).

	Proposed Method		Other Methods		
	SSA-DBN	GA-DBN	PSO-DBN	PSO-CNN	DBN
1	83.47	74.32	78.96	83.3	74.46
2	77.69	83.26	80.52	84.59	91.57
3	91.78	57.4	63.59	71.68	76.34
4	79.52	89.3	80.28	74.55	60.19
5	87.1	68.2	76.52	90.54	74.82
6	87	75.23	80.24	76.92	73.24
7	90.43	84.32	88.23	77.03	76.73
8	88.94	97.28	70.32	83.2	69.28
9	89.3	69.5	60.5	82.24	94.7
mean	**86.14**	**77.65**	**75.46**	**80.45**	**76.81**
std	**4.62**	**11.44**	**8.42**	**5.51**	**9.96**
SE	**1.54**	**3.81**	**2.80**	**1.83**	**3.32**

**Table 5 bioengineering-11-00030-t005:** SMR-BCI dataset classification accuracy (%) (AVG ± S.D).

	Proposed Method		Other Methods		
	SSA-DBN	GA-DBN	PSO-DBN	PSO-CNN	DBN
1	91.02	74.63	87.65	88.93	79.61
2	95.78	81.92	78.05	83.72	86.04
3	76.13	67.48	90.02	64.37	79.91
4	89.31	83.45	84.53	79.6	87.52
5	91.89	78.64	79.06	87.89	82.83
6	82.12	60.21	88.63	66.86	74.68
7	86.7	69.21	87.56	94	91.66
8	92.71	79.69	74.32	90.74	83.45
9	89.01	84.34	85.03	89.61	84.82
10	84.18	75.43	70.64	84.89	63.42
11	87.63	79.02	73.24	81.12	82.87
12	90.24	80.62	80.56	82.07	76.39
13	74.58	81.38	66.32	88.24	81.00
14	89.67	85.42	71.01	86.17	86.19
mean	**87.21**	**77.24**	**79.75**	**83.44**	**81.45**
std	**5.86**	**6.95**	**7.43**	**8.22**	**6.58**
SE	**1.57**	**1.86**	**1.99**	**2.20**	**1.76**

## Data Availability

The local data AO-MI that support the findings of this study are available on request from the corresponding author, and the data are not publicly available due to privacy or ethical restrictions. The BCI IV 2a dataset is available at https://www.bbci.de/competition/iv/ (accessed on 1 June 2023). The SMR-BCI dataset is available at https://min2net.github.io/docs/preprocessing/SMR-BCI/ (accessed on 18 June 2023).

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
