# Peer review of "Classification of EEG Signals Based on Sparrow Search Algorithm-Deep Belief Network for Brain-Computer Interface"

_bioengineering, 2023, doi:10.3390/bioengineering11010030_

Round 1
Reviewer 1 Report
Comments and Suggestions for Authors
This paper proposed an algorithm for classification of EEG signals for BCI. The idea is nice however, there are few recommendations that needs to be addressed:
1. writing should be improved. Continuity of the sentences is missing.
2. Abstract should be a single paragraph. This will make things easy for a reader.
3. More literature related to Automatic EEG signal classification should be added. You can cite:
a. Abbasi, S.F., Ahmad, J., Tahir, A., Awais, M., Chen, C., Irfan, M., Siddiqa, H.A., Waqas, A.B., Long, X., Yin, B. and Akbarzadeh, S., 2020. EEG-based neonatal sleep-wake classification using multilayer perceptron neural network. IEEE Access, 8, pp.183025-183034.
b. Abbasi, S.F., Jamil, H. and Chen, W., 2022. EEG-Based Neonatal Sleep Stage Classification Using Ensemble Learning. Computers, Materials & Continua, 70(3).
c. Abbasi, S.F., Abbas, A., Ahmad, I., Alshehri, M.S., Almakdi, S., Ghadi, Y.Y. and Ahmad, J., 2023. Automatic neonatal sleep stage classification: A comparative study. Heliyon.
d. Abbasi, S.F., Abbasi, Q.H., Saeed, F. and Alghamdi, N.S., 2023. A convolutional neural network-based decision support system for neonatal quiet sleep detection. Mathematical Biosciences and Engineering, 20(9), pp.17018-17036.
4. Introduction and related work section should be separated. There should be a separate "Related work" section.
5. Main contributions should be written in bullets at the end of Introduction section.
6. Structuring of the article should also be mentioned.
7. The experimentation includes human subjects. Approval number and authority should be added.
Reference related to the preprocessing technique is missing. You can cite:
Abbasi, S.F., Awais, M., Zhao, X. and Chen, W., 2019, June. Automatic denoising and artifact removal from neonatal EEG. In BIBE 2019; The Third International Conference on Biological Information and Biomedical Engineering (pp. 1-5). VDE.
8. More evaluation parameters should be added.
9. The author stated that they have used 5-fold cross validation to validate the overall dataset. The standard error is missing. It should be added.
10. Future directions should be briefly explained in the last paragraph of discussion section.
Comments on the Quality of English LanguageContinuity is missing.
Reviewer 2 Report
Comments and Suggestions for Authors
The authors of this paper present how the Sparrow Search Algorithm (SSA) can be used to optimise a Deep Belief Network (DBN) applied to motor-imagery recognition from EEG signals. They have used three datasets: their own dataset (8 subjects) and two public datasets, SMR-BCI dataset (14 subjects) and BCI competition IV dataset 2a (9 subjects). The authors show that SSA improves the classification accuracy of a DBN.
Mayor comments:
The manuscript is clearly written. However, there is some part of the manuscript that needs to be better descripted or completed.
In the point “2.1 Dataset” it should read “own dataset” or something similar, as further on it is written “2.2 Public datasets”. In the “2.1 Dataset”, it does not say how many subjects there are, this information is in line 385 in “3. Results”. There is no information about the subjects and there is no information about whether this study was approved by the relevant ethics committee (as well as its registration number).
The item “2.3.1. Sparrow Search Algorithm” is almost the same as the text that it is written in “Jiankai Xue & Bo Shen (2020) A novel swarm intelligence optimization approach: sparrow search algorithm, Systems Science & Control Engineering, 8:1, 22-34, DOI: 10.1080/21642583.2019.1708830“ but there is no reference to this article in this item, although the authors did refer to them in the “1. Introduction”. In any case, a better description could be made using a fuller, clearer and more intuitive explanation based on descriptions of the method by other authors.
In the “4. Discussion”, there is a figure showing the marginal spectrum of channels C3-, CZ and C4, which should be in the "Results" section. I recommend that these results be presented outside of "Discussion". In addition, it should be argued why these channels are selected. For example in the article “ Ruan, J., Wu, X., Zhou, B. et al. An Automatic Channel Selection Approach for ICA-Based Motor Imagery Brain Computer Interface. J Med Syst 42, 253 (2018). https://doi.org/10.1007/s10916-018-1106-3” there are some arguments.
If channels C3, Cz and CZ according to the authors allow to see differences between IM and AO, why has the classification of information been done with all channels? A comparison could be made to see the accuracy according to a selection of channels. This would make the classification method faster if the SSA-DBN could work just as well with fewer sensors.
At the end of "4. Discussion", the authors make some conclusions, it is better to limit the conclusions to "5. Conclusions". Discussion should therefore be re-written or merged with "3. Results".
This paper aims to demonstrate how SSA improves the classification accuracy of a DBN network applied to MI-BCI with EEG signals. SSA is a method that could be encompassed as a variant of the well-known PSO algorithms. Therefore, to make a correct comparison you should compare SSA with PSO and other algorithms such as GA, with the same DBN network and with the same database. On the other hand, the table shows the accuracy percentages of 8 persons (a very small number, to then make a comparative statistic as shown in Figure 7) and more metrics are recommended to better describe the scope of the proposed methodology.
Minor comments:
Line 114: to put references about “Curry8 software”.
Line 127: “AO” is used for the first time, it is supposed to be “ Action Observation”, it is advisable to define the abbreviation the first time it is used.
Line 144: It is said “Extraneous data …”, please use a more scientific or technical way of defining "Extraneous". How are they omitted? How are extraneous segments defined?
Line 197: It says “…Xij represents the position of the jth dimension of the first sparrow…”, do you mean “…the ith sparrow…”?
Line 239: It says “…bias terms a, b of the network…”, where is “a” in the equations?
Table 1 and Figure 7: is LE-SSVM and DE-LSSVM the same? Is PSD-CNN the same as PSO-CNN?
In Table1, to define abbreviations in the table. And briefly describe the classification algorithms used.
In the caption of Figure 7, define abbreviations.
In Figure 7, define the meaning of the stars (p=0.05, p=0.001, etc.), and in methods describes the statistical methods used to study the significance of the comparisons.
Figure 4: put the label “Hidden layers”.
Figure 7: in the y-axis, put label and vertical line of the y-axis and horizontal lines (e.g. dash-lines)
Reviewer 3 Report
Comments and Suggestions for Authors
Reviewer’s Report on the manuscript entitled:
Classification of EEG Signals based on SSA-DBN for Brain Computer Interface
The authors proposed a sparrow search algorithm optimized deep belief network to recognize the EEG features extracted by the empirical mode decomposition. They successfully showed the performance of their proposed model on two public and one private datasets. The manuscript is generally well-written, and the topic, method, and results are interesting. I have some comments for further improvement.
Title. Please avoid abbreviations in the title. Give the full name of SSA-DBN instead.
Introduction. The literature review can be improved and also the main contribution should be highlighted.
Line 36. The following review article on brain computer Interfaces for stroke neurorehabilitation can be added here: https://doi.org/10.3390/signals4010004
Line 70. Please define the acronym EEG. You defined EEG in line 97. All the abbreviations must be defined the first time they appear.
Line 70. EEG needs some introduction at least one paragraph. Please explain what EEG is and describe the type of noise in it, such as eyeblink and muscular artifacts as well as baseline removal paradigm that need to be removed prior to applications, such as emotion classification in the light of the following articles:
https://doi.org/10.3390/bioengineering10010054
https://doi.org/10.1016/j.bspc.2021.102741
Line 102. This sentence belongs to conclusions. Instead, here I suggest highlighting the main contributions of your work using bullet points or numbers.
Line 138, 139, etc. “data” is plural. Please check and correct the grammar everywhere.
Equation (4). Style issue. Please use larger size parentheses. Generally, the style and format of the equations should be checked. When you describe the parameters/variables of each equation, please ensure you use the same style/format for them.
Section 2.5. All these values and descriptions can be nicely listed inside a Table instead.
Figure 6, 7, 8. Please enlarge the font size of the texts and numbers.
Panel (b) in Figure 6. This is a frequency spectrum (periodogram) not a spectrogram. Please note that spectrogram is in time-frequency domain.
Caption of figure 7. Please rewrite the caption. Also please describe what those horizontal lines with stars on them mean.
Figure 8. It would be nicer to write which frequency ranges are for alpha and beta bands, etc. Also, what about higher frequency bands such as Gamma? Please elaborate.
In the discussion section please also elaborate on possible effect of artifacts and baseline on the EEG classification in the light of the last two articles that I suggested above.
Please also mention the limitations of your study at the end of conclusions and provide future direction.
Thank you for your contribution
Regards,
Comments on the Quality of English LanguageThere are many grammar/style/typo issues that must be checked and corrected.
Reviewer 4 Report
Comments and Suggestions for Authors
The following are my comments for improvement of this paper:
1. The authors state that the SMR-BCI dataset contains data from 14 subjects and the BCI IV 2a dataset contains data from 9 subjects. Please discuss why 14 subjects and 9 subjects should be considered as necessary and a sufficient number of subjects to uphold the effectiveness of the proposed system.
2. In Section 2.3.1, the mathematics related to the Sparrow Search Algorithm is presented. Please include a pseudocode or flowchart to explain the step-by-step working of this algorithm. To add to this, comment on the time complexity and space complexity of this algorithm.
3. The need or relevance of performing the research is not clearly written. The authors state about the “quality of life for individuals with disabilities” but the discussion is presented in a very abstract manner and no specific needs of the target population are highlighted. Some potential needs of individuals with disabilities as well as elderly people could be assistance in Activities of Daily Living, Navigation, etc. Consider reviewing these papers - https://doi.org/10.3390/info12030114 and https://doi.org/10.1109/ICTAS.2019.8703637 to present a list of potential needs of the target population that this work could address.
4. Several fact-based statements throughout the paper are missing supporting references. For instance, this statement – “The advancements in BCI research not only contribute to the development of artificial intelligence but also hold great potential for enhancing healthcare and promoting the integration of human and machine capabilities.” should have a supporting reference.
5. A comparison with prior works is missing: Please include a comparative study (qualitative and quantitative) with prior works in this field to highlight the novelty of this work.
6. The limitations of the work should be clearly stated.
Round 2
Reviewer 1 Report
Comments and Suggestions for Authors
All comments have been addressed!!!
Author Response
Thank you very much for your hard work on this manuscript. Your valuable feedback has been of great help to this article.
Reviewer 2 Report
Comments and Suggestions for Authors
The authors have responded adequately to the issues raised.
Author Response
Thank you for your correction of this manuscript. We express our sincerest gratitude for your efforts.
Reviewer 3 Report
Comments and Suggestions for Authors
Dear authors,
Thank you for the revisions, but I still see that some of my comments are not addressed properly. Please see below and carefully address them. Please also avoid using the red color for highlights. Please use a light yellow color instead:
Regarding my comment #3 in lines 52 and 57 of your revised draft, please also review my suggested two references that are discuss the importance of de-noising EEG signals before any classifications.
Regarding my comment #4. The main contributions should talk about what you intend to do not providing the results in it. For example, lines 117-121 you provided the numerical results. Instead of lines 117-121 you need to say something like "To compare the proposed method with three baseline methods".
Regarding my comment 6: Equations (3)-(5). Please remove the large parentheses. Instead use one large curly bracket { after =.
Regarding my comment 8: Figure 6. The x-axis label. Please use a consistent fort size. Panel (a) x-axis is Time (s). Please enlarge "s". Also panel (c) should be Time (s) instead of Time (sec). Please be consistent with the font size and units. Please check other figures for similar issues.
Figure 7 has tiny font size for the axis values. Please enlarge.
Regarding my comment 7. Thank you for adding Tables 1 and 2. But I do not see anywhere in lines 382-395 that you refer to Table 1. Please note that all the tables and figures must be referred to in the manuscript.
Please check equations (24) and (25) to ensure their correctness.
Comment 12 is not properly discussed in line of the suggested articles.
Line 592. The limitation and future work section should be before the conclusion section.
Too many newly added references by the same authors [4],[7],[8],[9],[33].
Please carefully proofread the manuscript.
Thank you!
Comments on the Quality of English LanguageThere are grammar/punctuation/style issues that should be checked and corrected.
Reviewer 4 Report
Comments and Suggestions for Authors
The authors have revised their paper as per all my comments and feedback. I do not have any additional comments at this point. I recommend the publication of this paper in its current form.
Author Response
Thank you very much for your recognition and support of this manuscript.